# Co-Occurrence of Congenital Aniridia Due to Nonsense *PAX6* Variant p.(Cys94*) and Chromosome 21 Trisomy in the Same Patient

**DOI:** 10.3390/ijms242115527

**Published:** 2023-10-24

**Authors:** Tatyana A. Vasilyeva, Natella V. Sukhanova, Andrey V. Marakhonov, Natalia Yu. Kuzina, Nadezhda V. Shilova, Vitaly V. Kadyshev, Sergey I. Kutsev, Rena A. Zinchenko

**Affiliations:** Research Centre for Medical Genetics, 115522 Moscow, Russia; vasilyeva_debrie@mail.ru (T.A.V.); natelasukhanova@gmail.com (N.V.S.); ryzhaja@bk.ru (N.Y.K.); nvsh05@mail.ru (N.V.S.); vvh.kad@gmail.com (V.V.K.); kutsev@mail.ru (S.I.K.); renazinchenko@mail.ru (R.A.Z.)

**Keywords:** rare diseases, co-occurrence, complex phenotype, *PAX6* nonsense variant, congenital aniridia, trisomy 21, Down syndrome

## Abstract

This study aims to present a clinical case involving the unique co-occurrence of congenital aniridia and Down syndrome in a young girl and to analyze the combined impact of these conditions on the patient’s phenotype. The investigation involved comprehensive pediatric and ophthalmological examinations alongside karyotyping and Sanger sequencing of the *PAX6* gene. The patient exhibited distinctive features associated with both congenital aniridia and Down syndrome, suggesting a potential exacerbation of their effects. Cytogenetic and molecular genetic analysis revealed the presence of trisomy 21 and a known pathogenic nonsense variant in exon 6 of the *PAX6* gene (c.282C>A, p.(Cys94*)) corresponding to the paired domain of the protein. The observation of these two hereditary anomalies offers valuable insights into the molecular pathogenetic mechanisms underlying each condition. Additionally, it provides a basis for a more nuanced prognosis of the complex disease course in this patient. This case underscores the importance of considering interactions between different genetic disorders in clinical assessments and treatment planning.

## 1. Introduction

Congenital aniridia (AN, OMIM #106210) is an autosomal dominant condition in a great majority of the cases resulting from heterozygous intragenic *PAX6* pathogenic variants, primarily loss-of-function mutations or 11p13 chromosome deletions. Rarely, variants in the *FOXC1*, *PITX2, PITX3*, and other genes may also be a cause of AN [1,2]. AN is characterized by iris absence, foveal hypoplasia, and other ocular anomalies. AN may also manifest extraneous systemic effects, such as impacts on the nervous, endocrine, and immune systems, suggesting potential syndromic manifestations associated with *PAX6*-related AN [3]. In the Russian Federation, AN occurs at a rate of 1 per 98,943 individuals [Vasilyeva, in press], while Orphanet reports a prevalence of 1 per 76,335 [4]. In the world, AN prevalence is 1:40,000 to 1:100,000 [1].

Down syndrome (DS, OMIM #190685) presents a common chromosome syndrome which is caused by partial or complete chromosome 21 trisomy. DS is characterized by mental retardation, distinct facial features, and potential systemic involvement. DS patients often have neurologic problems; eye abnormalities and decreased visual acuity; hearing loss; cardiac anomalies and related pulmonary disorders; renal anomalies; thyroid dysfunction; diabetes; immunologic deficit due to a decreased number of T and B lymphocytes; myeloid leukemia; growth delay; obesity; and others [5,6,7,8]. DS newborns exhibit a significantly higher incidence of congenital malformations, including microphthalmia and congenital cataracts [9]. Ocular anomalies associated with DS, in addition to characteristic palpebral fissure slanting and epicanthal folds, encompass strabismus, hyperopia, myopia, astigmatism, iris thinning and hypoplasia, Brushfield spots, retinal anomalies, nystagmus, and glaucoma [8]. In the world population, DS prevalence is assessed as 1 per 700 people [8].

Ocular anomalies in DS and AN can overlap; strabismus, refractive errors, cataract, and nystagmus occur very often with both syndromes, though the frequencies vary. Other affections of the eye can differ; special cornea, iris, optic nerve, and retina clinical features characterize each of the two disorders. The thinning and conical shape of the cornea, Brushfield spots composed by hyperplasic iris stromal connective tissue, optic nerve head elevation and its vascularization, and central macula thickness can characterize DS, while complete or partial iris absence or its severe hypoplasia, cornea thickening, limbal deficiency, keratopathy, and optic nerve and foveal hypoplasia are the features of an aniridic eye [8,10]. In this report, we present a genetically confirmed case of *PAX6*-associated AN co-occurring with DS in a single patient, a girl who is currently 4 years old. We also discuss potential interactions between these two distinct disorders, as DS and AN can affect the same organs and systems: brain, eye, ear, and others.

## 2. Case Presentation

The patient, a female infant and the second child in a nonconsanguineous family, was born at 38 weeks of gestation via natural delivery, weighing 3600 g and measuring 54 cm in length. The first child, the proband’s 19-year-old elder brother, and their parents are healthy. There was no hazard exposure to the mother during the pregnancy. The mother underwent the regular prenatal screening, which showed an increased risk of chromosomal pathology. Nevertheless, the family decided not to terminate the ongoing pregnancy because of the medical reason.

Apgar scores were 7 at 1 min and 8 at 5 min. In the first week after birth, the baby received neonatal ventilator treatment for respiratory syndrome in the intensive care unit. Postnatal karyotyping confirmed trisomy of chromosome 21.

Upon discharge, the infant was diagnosed with Down syndrome, congenital bilateral complete aniridia, congenital bilateral glaucoma, unilateral right megaureter, and respiratory distress syndrome.

At 3–4 months of age, the patient began tracking toys. By 7 months, she gained control of her head. Rolling from tummy to side became evident at 5 months. At 1 year and 3 months, she could sit unsteadily, though she could not yet pull herself up. Starting at 10 months, she demonstrated the ability to wiggle while in a standing position, supporting herself on her arms and knees.

At 3 months of age, brain neurosonography showed no pathology.

At 6 months of age, body ultrasonography detected a unilateral right megaureter.

At 12 months of age, heart ultrasonography revealed persistent ductus arteriosus and an atrial septal defect.

At 4 years old, ultrasonography of the eyeballs revealed the following: In the right eye (OD), there was aniridia, with an anterior chamber depth of 3.5 mm and an axial length of 22.7 mm. Hyperechogenic inclusions were observed in the vitreous body, primarily in its posterior region. The lens exhibited normal topography, with occasional slight subcapsular tissue compactions. The retrobulbar space remained intact.

In the left eye (OS), aniridia was also present, along with an anterior chamber depth of 3.8 mm and an axial length of 24.2 mm. Multiple hyperechogenic inclusions were noted in the vitreous body, including both point and linear formations. The lens displayed normal topography but with pronounced subcapsular tissue compactions. The retrobulbar space was intact.

Latency auditory evoked potentials indicated unilateral sensorineural hearing loss, with normal hearing in the right ear.

The patient’s medical history encompasses various aspects.

In terms of neurology, she presents with facial dysmorphism, including a depressed nasal bridge, hypertelorism, antimongoloid slant, and eyelid epicanthal folds. Additionally, there is an asymmetry in the eyes due to left eye buphthalmos. Neurologically, she exhibits horizontal nystagmus, heterotropia, high amblyopia, open mouth, macroglossia, hypersalivation, myotonic syndrome, reduced tendon reflexes, and developmental delay (see Appendix A and Figure 1 for visual reference). Furthermore, she experiences unilateral hearing loss.

Her ophthalmologic history is marked by congenital eye pathology, specifically bilateral complete aniridia, congenital bilateral partially drug-compensated glaucoma, cornea opacity, keratopathy, optic nerve and foveal hypoplasia, high amblyopia, and mild myopia (see Appendix A and Figure 1 for visual reference).

In the realm of urology, she was diagnosed with unilateral right megaureter, necessitating planned operative treatment.

The endocrinologic history reveals an elevated thyrotropic hormone level (TTH).

She also has an otolaryngologic diagnosis of laryngomalacia.

Orthopedically, she presents with postural kyphosis abnormality, weakened back muscles, valgus deformity in the right foot, and varus deformity in the left foot. Additionally, she was diagnosed with connective tissue dysplasia.

As she has undergone several years of examination at RCMG, we have maintained a dynamic record of her disease progression from birth until the present. A comprehensive overview is provided in Appendix A.

The initial cytogenetic study revealed a karyotype of 47,XX,+21 (Figure 2a). Subsequent MLPA analysis showed no 11p13 chromosome misbalance in either the patient or her parents.

Further genetic investigation through *PAX6* gene screening identified a known heterozygous variant NM_000280.4:c.282C>A, p.(Cys94*), in the patient (Figure 2b). The variant c.282C>A leads to a premature stop codon formation, which is registered in the Human Genome Mutation Database (HGMD reference number CM205358), that was found and classified as a causative pathogenic variant in a large cohort of Chinese patients with AN in 2020 [11]. Notably, this variant was not present in her parents. The NM_000280.4:c.282C>A variant leads to the formation of a premature stop codon, p.(Cys94*). It is crucial to note that this variant was not found in the control sample of healthy individuals (gnomAD) and occurred de novo in the patient. Previously, this variant was identified in another patient with AN [11]. Following assessment according to the ACMG recommendations, it was classified as a pathogenic variant, taking into account pathogenicity criteria PM2, PVS1, PS2, and PP5 [12].

## 3. Discussion

The parents of the patient, a 4-year-old girl and the second child in the family, sought diagnosis at RCMG. Initially, there were suspicions of both Down syndrome (DS) and congenital aniridia (AN), which were later confirmed through cytogenetic and molecular genetic testing. The patient’s karyotype revealed 47,XX,+21 (Figure 2), while her parents displayed normal karyotypes (not shown). Sanger sequencing identified a de novo pathogenic sequence variant NM_000280.4:c.282C>A, p.(Cys94*) in the *PAX6* gene in a heterozygous state (Figure 2). The *PAX6* gene encodes one of the key embryonic transcription factors, a master regulator of eye and central nervous system morphogenesis and maintenance. During embryogenesis, *PAX6* is expressed in almost all eye structures. This synchronizes development of different eye tissues and provides the development of a mature and functioning eye. Defined in the patient, the genetic variant leads to a premature stop codon formation and to a loss of function of one *PAX6* allele, in other words, to *PAX6* haploinsufficiency and AN phenotype.

The patient manifests a combination of features of both DS and AN [1,6]. These include complete bilateral aniridia, nystagmus, strabismus, congenital glaucoma, buphthalmos, AN-related keratopathy at stage 2 with a vascularized corneal pannus measuring approximately 3 mm, and cataract in one eye (the lens status in the other eye cannot be assessed). Additionally, she exhibits macular hypoplasia and congenital optic nerve head pathology, specifically glaucomatous excavation. Facial features consistent with DS are also observed, alongside developmental delay, unilateral hearing loss, myotonic syndrome, reduced tendon reflexes, postural kyphosis abnormality, persistent ductus arteriosus, atrial septal defect in her heart, and a history of operation on a unilateral megaureter.

DS arises from trisomy of chromosome 21 and is among the most prevalent hereditary disorders, occurring at a frequency of 1 per 500 to 1000 individuals [4]. Given its high prevalence, DS often co-occurs with various hereditary anomalies, chromosomal syndromes, or Mendelian disorders in affected individuals [13]. The simultaneous occurrence of AN and DS in a single patient is theoretically rare, estimated at 1 per 76,335,000 to 98,943,000. In the Russian Federation population of approximately 146 million, this corresponds to one to two cases; globally, it may correspond to several dozen.

Previously, the unique combination of DS and AN was reported only twice. One instance involved a 4.5-month-old boy who presented as a sporadic case of both AN and DS [14], presumed to be a rare coincidental event (*PAX6* sequence analysis was not performed) (Patient 2 in Table 1).

The second reported case involved a boy with complex brain anomalies and extreme microphthalmia [15] (Patient 3 in Table 1). This individual was identified as a compound heterozygote for two mutations in the *PAX6* gene: a missense variant c.112C>T, p.(Arg38Trp), inherited from the father, and a nonsense variant c.718C>T, p.(Arg240*), inherited from the mother. Additionally, he exhibited de novo chromosome 21 trisomy. Interestingly, the father displayed a cataract, subtle iris hypoplasia, corectopia, microcornea, and hearing loss, with an extensive family history of these conditions. The mother had complete aniridia, which also had a familial history [15]. The paternal missense variant p.(Arg38Trp) was previously associated with microphthalmia, while the maternal p.(Arg240*) variant was linked to aniridia [16,17].

Detailed clinical profiles for each individual are outlined and compared in Table 1.

DS may exacerbate co-occurring conditions if the affected genes function within the same or closely related pathways [18,19,20,21,22]. The interplay could aggravate (i) eye phenotype, (ii) bone/growth phenotype, (iii) ear, because of sensor (AN) craniofacial abnormalities (AN and DS), (iv) neurologic phenotype because of poor neurodevelopment and an increased risk of autistic spectrum and attention deficit disorder with AN and DS, and (v) diabetes due to pancreatic function deficit with AN and due to autoimmune consequences with DS [6]. This interplay may also impact patient survival, as illustrated by a reported case of a patient with trisomy 21 and a potentially lethal combination of two biallelic *PAX6* mutations who survived, while their elder brother, carrying only two biallelic *PAX6* mutations, succumbed shortly after birth [15].

While the report on the 4.5-month-old boy (Patient 2 in Table 1) is relatively brief and the case may be underexamined and underestimated, it is noteworthy that neither his eye phenotype nor his developmental delays nor additional system affections appear to be severe [14].

A range of severity of Down syndrome can be observed in different patients, which is supposed to be related to the length of the partial triplication of chromosome 21. Mosaic for trisomy 21 and partial 21 trisomic patients present the less severe DS features [23]. Our patient has a complete 21 trisomy and should have a fully expressed DS phenotype. Nevertheless, the girl described here seems to exhibit a mild phenotype, possibly except for the presence of congenital glaucoma (Patient 1 in Table 1). Conversely, the third case, with a complex phenotype of DS plus AN, is the most severe due to compound heterozygosity for two pathogenic *PAX6* variants in addition to chromosome 21 trisomy. This case potentially illustrates the dosage effect of *PAX6*, where severe phenotypes manifest when both *PAX6* alleles are affected, if the individual is viable at all [24].

A complex phenotype resulting from the interaction of DS and AN, which can itself be considered a syndrome, is anticipated to present a very severe clinical picture in the patient. Other organs and systems may also be affected, as observed in about half of AN cases based on our previous study of a large AN patient cohort [25].

Both DS and AN are recognized as multisystem developmental disorders. They can impact not only eye development but also the development and functioning of other organs and systems. For example, features like glaucoma, keratopathy, nystagmus, amblyopia, as well as hearing loss, growth deficiency, and central nervous system involvement can be characteristics of both conditions, potentially exacerbating each other [26,27,28]. Despite the presence of all these features in our patient, her condition is assessed as moderately severe.

Recently, the patient has shown active engagement in communication with her family members, other children, and doctors. She is able to play and respond to requests. She participates in comprehensive developmental classes, including sessions with a defectologist, neuropsychologist, and sensory integration specialist. The persistence and support provided by her parents in her socialization and healthcare are invaluable for both her and themselves.

Though we do not observe any aggravation of at least neurologic and eye affections due to the combination of the two severe congenital pathologies, in every case of such a rare combination, an individual and multidisciplinary approach is needed.

Biological mechanisms underlying the DS phenotype are not well understood but could include a dosage effect of overexpressed genes due to the triplication of chromosome 21 and its consequences and the genome-wide disruption of expression regulation and all consequences. However, no link is established between dysregulated genes and specific clinical manifestations of DS, so genes on the trisomic chromosome could be a target for drug development for DS. Animal trisomic models of DS demonstrate learning deficits and altered synaptic function in the hippocampus and the improving effect of some of pharmaceutical agents’ usage [23].

The most studied feature of DS, early-developed neurodegeneration and dementia, was proposed to be the consequence of the amyloid-β precursor (APP) locus triplication due to 21 chromosome trisomy. The same mechanisms for dementia in DS and Alzheimer’s disease were suggested [29]. Recently, the amyloid-β molecular signature in DS was found to be distinct from that in Alzheimer’s disease; thus, mechanisms and therapeutic approaches should also be different [30].

We could provide only suggested mechanisms of both conditions’ pathogenesis and their interactions, as there is still a lack of knowledge on the topic, e.g., neurodegenerative processes in brain cells of the patients with *PAX6* pathogenic variants were proposed. *PAX6* was shown to regulate genes essential for immunological surveillance and energy metabolism in the brain that change with aging [31]. Concerning brain tissues, the phenotype associated with DS observed in trisomy 21 mitochondrial dysfunction could be a mechanism. Altered mitochondrial energy metabolism and oxidative stress could lead to intellectual disability and Alzheimer’s disease in DS patients [30].

Mechanisms of neurological phenotypes in patients with AN were also only proposed; they could involve functional and structural neuroplasticity deficits [32]. Little is known about exactly how they work in the developing humane brain due to the brain’s complexity. We can probably expect some aggravation of a DS-linked cognitive deficit with that caused by *PAX6* function haploinsufficiency.

Practical recommendations and vigilance are as follows: Long-term follow-up will enable a detailed longitudinal analysis of the patient’s ophthalmic and systemic conditions. The severe cornea opacity, keratopathy, and glaucoma, which are characteristic features of *PAX6*-associated aniridia, necessitate urgent attention from parents and ophthalmologists [33,34,35]. Given signs of connective tissue dysplasia and contraindications for using antimetabolites to suppress postoperative scarring, glaucoma surgery is not advisable due to an extremely poor prognosis for surgical treatment. However, this feature significantly impairs the patient’s quality of life and poses a constant and significant threat to her health. Therefore, a crucial specialist’s note is to exercise caution with eye operations, which could potentially lead to blindness, particularly due to the risk of aniridic fibrosis syndrome [33,36]. For now, the only feasible solution for elevated intraocular pressure is the use of antiglaucoma medications. Additionally, trisomy 21 necessitates heightened awareness regarding immune system deficiency [37,38]. Also, early dementia and neurodegenerative processes threaten DS patients.

## 4. Conclusions

The co-occurrence of DS and AN in this patient results in a phenotype that encompasses characteristics of both syndromes, with some shared features that do not seem to exacerbate each other.

The reason for the observed complex condition is the random co-occurrence of an intragenic *PAX6* pathogenic variant and a chromosome anomaly, 21 trisomy, as well as the viability of the child with this genetic combination. We supposed that the phenomenon could be explained by the involvement of totally different pathways as well as possibly increased organism survival ability due to increased challenge.

This case is indicative of the de novo emergence of these two disorders, leading to a relatively mild phenotype. However, it is important for the parents to remain vigilant regarding potential developments of keratopathy, glaucoma, and immunodeficiency in the patient.

We found and discussed only the main causative genetic variants, which are necessary and sufficient conditions for the observed phenotype; additional findings could be present or absent, though the individual’s genetic background could more or less influence the expression of causative variants.

The co-occurrence of AN and DS could be a subject for investigation of complex genetic interactions between different loci. We suggest that such coappearance cases might be excellent examples of phenotypic presentations of genetic interaction on genic and chromosomal levels.

## Figures and Tables

**Figure 1 ijms-24-15527-f001:**
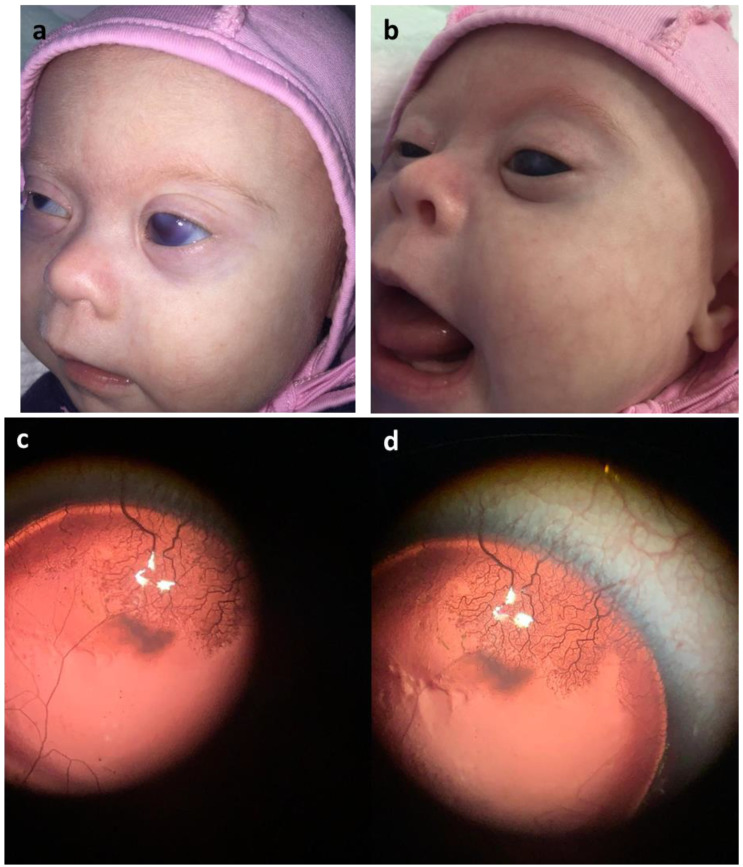
(**a**,**b**) OS buphthalmos and congenital glaucoma. Corneal opacity. (**c**,**d**) Complete aniridia (no iris remnant tissue is visible at slit lamp examination, without gonioscopy). The pannus invades the central cornea, typically covering the entire cornea with vessels.

**Figure 2 ijms-24-15527-f002:**
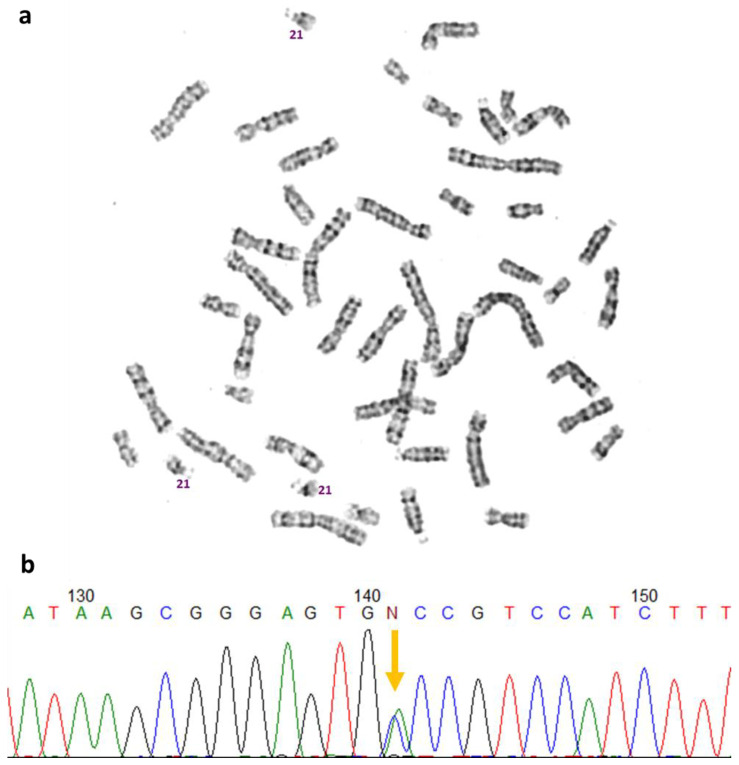
Genetic examination of the patient: (**a**) karyotype showed 47,XX,+21; (**b**) Sanger sequencing revealed NM_000280.4(*PAX6*):c.282C>A, p.(Cys94*), variant in a heterozygous state (showed with arrow).

**Table 1 ijms-24-15527-t001:** The comparison of clinical signs of the three known patients with complex DS and AN.

Patients	Patient 1	Patient 2	Patient 3
Age, gender	A 4-year-old girl	A 4.5-month-old boy	A 4-year-old boy
Reference	(this study)	[14]	[15]
Karyotype	47,XX,+21	47,XY,+21	47,XY,+21
*PAX6* genotype	c.[282C>A];[=]	not tested	c.[112C>T];[718C>T]
Facial features consistent with diagnosis of trisomy 21	Facial dysmorphism, depressed nasal bridge, hypertelorism, antimongoloid slant and eyelid epicanthal folds, asymmetric eyes due to the left eyeball buphthalmos, open mouth, macroglossia, hypersalivation	Facial dysmorphism, depressed nasal bridge, hypertelorism, antimongoloid slant and eyelid epicanthal folds	Facial dysmorphism, depressed nasal bridge, hypertelorism, antimongoloid slant and eyelid epicanthal folds, asymmetric microphthalmia, open mouth, macroglossia, hypersalivation
Neurologic status	Myotonic syndrome, reduced tendon reflexes, postural kyphosis abnormality, weakened back muscles, valgus right foot and varus left foot deformities	not tested	Extreme microcephaly, a smooth philtrum
Developmental delay	Developmental delay	not tested	Severe developmental delay
Brain structure	Normal brain structure	not tested	Complex structural brain anomaly, corpus callosum agenesis, midline interhemispheric cyst, hypoplastic pons and vermis (absent inferiorly), dysplastic tectum, pituitary and hypothalamic hypoplasia, and a globular (though not fused) basal ganglia
Eyes	Congenital bilateral complete aniridia	Congenital bilateral complete aniridia	Bilateral extreme microphthalmia without visual function
	Congenital bilateral glaucoma	No glaucoma, normal eye pressure	No data
	Clear lenses with first signs of developing opacities	Clear lenses	No data
	Vitreous bodies with first signs of degeneration	Normal vitreous bodies	No data
	Keratopathy OSCornea opacity OD	Clear cornea without vascularization or other signs of keratopathy	No data
	Normal anterior chambers	Normal anterior chambers	No data
	Horizontal nystagmus	Horizontal nystagmus	No data
	Mild hypoplasia of the optic nerve disks	Mild hypoplasia of the optic nerve disks	No data
	Foveal hypoplasia	Foveal hypoplasia	No data
	Albinotic color of both fundi	Subalbinotic changes in both fundi	No data
	Strabismus	No strabismus	No data
	Myopia	not tested	No data
Hearing	Unilateral hearing loss	Normal hearing	No data
Heart	Persistent ductus arteriosus, atrial septal defect	Ventricular septal defect, aortic coarctation	No data
Airway anomalies	Laryngomalacia, breath failure on the first week after birth	Bronchopulmonary dysplasia, breath failure	Choanal atresia
Renal anomalies	Megaureter	No anomalies	Renal dysplasia with recurrent urinary tract infections
Endocrine status	Increased TTH level	not tested	Central hypothyroidism Gonadotropin deficiency Cryptorchidism Secondary adrenal insufficiency Insulin-dependent diabetes mellitus without pancreatic anomalies

## Data Availability

The datasets used and/or analyzed during the current study are available from the corresponding author upon reasonable request.

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
