# Peer review of "Co-Occurrence of Congenital Aniridia Due to Nonsense *PAX6* Variant p.(Cys94*) and Chromosome 21 Trisomy in the Same Patient"

_ijms, 2023, doi:10.3390/ijms242115527_

Round 1
Reviewer 1 Report
The discussed work constitutes a significant contribution to the field of genetics and molecular medicine, investigating the rare phenomenon of the co-occurrence of congenital aniridia with a mutation in the PAX6 gene and chromosome 21 trisomy in three patients. PAX6 is a gene responsible for eye development, and its mutations are known as a cause of various visual impairments, including aniridia. When combined with chromosome 21 trisomy, typically associated with Down syndrome, this presents a unique case that requires in-depth understanding and analysis.However, there are some points to consider in the context of this manuscript.
1. Abstract:
-
It would be beneficial if the authors could provide more information about the molecular research results and their implications. Is the p.(Cys94*) mutation already known to be pathogenic in the scientific literature, or is it a novel mutation that requires further functional studies?
-
Did the authors consider the possibility of co-occurring other genetic changes in this case? Have they conducted investigations into other mutations related to congenital aniridia or Down syndrome?
-
It would also be worthwhile to discuss specific clinical implications for this patient and how it may impact her treatment and medical care.
- Introduction:
-
Introduction: The introduction is well-written and provides a clear overview of two rare conditions - congenital aniridia and Down syndrome. However, it is worth noting that in the case of congenital aniridia, mutations in the PAX6 gene are the primary cause, but other genetic causes may exist and should be considered.
-
Epidemiological Data: Providing information about the prevalence of these conditions in Russia and from Orphanet is valuable, but it would be even more informative if the authors could provide epidemiological data on a global scale to better understand the prevalence of these disorders worldwide.
-
Clinical Characteristics: The description of the characteristic features of both disorders is concise but adequate. However, it may be worth considering adding a more detailed description of the clinical symptoms in congenital aniridia and Down syndrome to help the reader better understand these conditions.
-
Ocular Anomalies: The description of ocular anomalies associated with Down syndrome is detailed and well-done, but it might be valuable to provide more information about the typical clinical features in congenital aniridia to allow readers to compare both conditions.
-
Study Objective: The study objective is clear - to present a genetically confirmed case of PAX6-associated congenital aniridia co-occurring with Down syndrome in a single patient and to explore potential interactions between these two distinct disorders. This is well-defined and serves as motivation for further reading.
-
Overall, the text is well-written and conveys important information. Consideration could be given to providing more detailed clinical symptom data for both conditions and expanding on the epidemiological information.
- 2. Case presentation:
-
Comments on the Case Presentation:
-
Patient Presentation: The description of the patient is detailed and includes important information such as her age, birth conditions, Apgar scores, and the confirmation of trisomy 21 through karyotyping. This is important for providing a comprehensive overview of the patient's situation from birth.
-
Child's Development: The description of the child's development focuses on several key developmental milestones, providing insights into her progress and achievements. It's worth noting that these details are useful for assessing her condition and treatment.
-
Medical Tests: The authors provide results from various tests, such as neurosonography, eye ultrasound, hearing tests, cardiac ultrasound, and others. This is important for gaining a thorough understanding of the extent of the patient's conditions.
-
Eye Findings: The description of eye findings is accurate and includes information about both eyes, which is crucial for evaluating her eye health and vision.
-
Clinical Aspects: The authors provide an extensive description of various clinical aspects of the patient, including neurological, ophthalmological, urological, endocrinological, otolaryngological, and orthopedic aspects. This is important to consider the complexity of her health condition.
-
Medical History: The description of the medical history is comprehensive and includes information on various aspects of the patient's health. This is valuable information for doctors conducting further research and treatment.
-
Genetic Test Results: The authors conducted genetic tests and identified a heterozygous variant NM_000280.4:c.282C>A, p.(Cys94*) in the PAX6 gene. They also describe that this variant is not present in the patient's parents and is classified as pathogenic. This is important information that can help understand the cause of her condition.
-
Disease Progression: The description of the disease progression of the patient is comprehensive and provides insights into various aspects of her health from birth to the present.
- Discussion:
-
Interactions Between DS and AN: In the discussion, the authors address the combination of features associated with Down syndrome (DS) and congenital aniridia (AN). They explain that both conditions may influence each other, but they do not provide a deeper analysis of the molecular or biological mechanisms that might underlie these interactions. It would be valuable to expand this section to understand why these two conditions might co-occur and how they may mutually affect each other.
-
Case Analysis: The authors present a comparison of three cases of patients with the combination of DS and AN. However, a more in-depth clinical analysis of each case and their differences and similarities is lacking. Adding a more detailed assessment of the symptoms and test results for each patient would help better understand the distinctions between them.
-
Potential Pathogenesis: The discussion does not provide an explanation for the pathogenesis of this unique combination of DS and AN. While the authors mention the possibility of genes related to these conditions influencing each other, they do not offer specific information on this topic. This is an important aspect that could be further developed to better understand the reasons for the occurrence of these two conditions in one patient.
-
Therapeutic Perspectives: Consider adding information about therapeutic perspectives for the patient. Are there any innovative therapies or approaches that could help manage her health condition? This could be valuable for other doctors and researchers dealing with similar cases.
-
Summary: The discussion lacks a summary of the main conclusions and messages. Consider adding a brief summary that highlights the key aspects of the discussed case and the insights gained from this analysis.
Overall, the discussion is interesting and contains important information about the patient's case, but it could be further expanded and deepened. Particularly, understanding the pathogenesis of the co-occurrence of these two conditions in one individual is crucial.
5. Conclusions:
-
Clinical Assessment: In the conclusions, the authors summarize that the patient has a phenotype that encompasses characteristics of both Down syndrome (DS) and congenital aniridia (AN), with some shared features that do not seem to exacerbate each other. This is a good overall assessment, but it would be beneficial to provide a more detailed description of which features are common and which are different between these two conditions.
-
De Novo Emergence: The authors note that these two conditions appeared de novo in the patient, resulting in a relatively mild phenotype. However, they do not provide a deeper analysis of why and how these two conditions emerged de novo in this specific patient. Exploring the reasons behind this occurrence would be valuable.
-
Guidance for Parents: The authors mention that parents should remain vigilant regarding potential complications related to the eyes and the immune system in the patient. It would be helpful to add specific recommendations or precautions that parents should take in terms of monitoring and managing these potential complications.
-
Consideration for Future Research: In the conclusions, it might also be worth considering what future research directions could be useful in better understanding this unique case and its clinical implications. Are there genetic or molecular aspects worth delving into in the context of this case?
-
Summary: Consider adding a brief summary that emphasizes the key findings from the analysis of this case and the potential implications for future patient care.
Overall, the "Conclusions" section is good but could be enhanced with more detailed information about common and differing features and recommendations for parents regarding monitoring the patient.
The overall quality of English language usage in the manuscript is quite good, with clear and understandable writing. However, there are some areas where improvement is needed:
-
Grammar and Sentence Structure: There are instances where sentence structure could be more concise and grammatically correct. For example, in some sentences, there is an unnecessary use of passive voice or complex phrasing that can be simplified for clarity.
-
Clarity and Precision: While the text is generally clear, there are sections where the meaning of certain sentences or phrases could be clarified for better comprehension. Some sentences appear slightly ambiguous or wordy.
-
Terminology and Abbreviations: Ensure that medical and scientific terminology is consistently used and properly defined, especially when introducing abbreviations. Readers should be able to easily understand all specialized terms.
-
Consistency: Maintain consistency in formatting, punctuation, and style throughout the manuscript. This includes consistent use of tense (past or present), punctuation (e.g., the use of commas), and formatting (e.g., italics for scientific names).
-
Citation and Reference Style: Verify that the citation and reference style adheres to the specific guidelines or standards required by the journal or publication.
-
Proofreading: Conduct a thorough proofreading to eliminate typographical errors, such as missing or duplicated words, punctuation mistakes, and formatting issues.
Overall, the manuscript is well-written, but careful editing for grammar, clarity, and consistency will further enhance the quality of the English language usage.
Author Response
Reviewer 1
The discussed work constitutes a significant contribution to the field of genetics and molecular medicine, investigating the rare phenomenon of the co-occurrence of congenital aniridia with a mutation in the PAX6 gene and chromosome 21 trisomy in three patients. PAX6 is a gene responsible for eye development, and its mutations are known as a cause of various visual impairments, including aniridia. When combined with chromosome 21 trisomy, typically associated with Down syndrome, this presents a unique case that requires in-depth understanding and analysis.However, there are some points to consider in the context of this manuscript.
- Abstract:
- It would be beneficial if the authors could provide more information about the molecular research results and their implications. Is the p.(Cys94*) mutation already known to be pathogenic in the scientific literature, or is it a novel mutation that requires further functional studies?
A1: The variant NM_000280.4(PAX6):c.282C>A leads to a premature stop codon formation p.(Cys94*), that is registered in Human Genome Mutation Database with reference number CM205358, that was found and classified as causative pathogenic variant in a Chinese cohort of AN patients in 2020 (Pubmed: 32214788). We added a corresponding information to the text, thank you. (Lines 15,120-124, 148-154)
- Did the authors consider the possibility of co-occurring other genetic changes in this case? Have they conducted investigations into other mutations related to congenital aniridia or Down syndrome?
A2: Of course, the possibility of several other additional genetic changes co-occurring in this case could and should be assessed. We have found and discussed only main causative genetic variants which are necessary and sufficient, as well as explainable conditions for the observed phenotype, additional findings could be present and could be absent, though individual genetic background always more or less influences expression of causative mutations. Thank you, we added that part to the Conclusion section (Lines 287-290).
- It would also be worthwhile to discuss specific clinical implications for this patient and how it may impact her treatment and medical care.
A3: Lines 260-273 are devoted to the discussion of the issue.
Introduction:
- Introduction: The introduction is well-written and provides a clear overview of two rare conditions - congenital aniridia and Down syndrome. However, it is worth noting that in the case of congenital aniridia, mutations in the PAX6 gene are the primary cause, but other genetic causes may exist and should be considered.
A4: Thank you, that’s right. We added the sentence into the introduction section. (Lines 25-28) PAX6 pathogenic variants, intragenic and copy number variants, previously 11p13 deletions, are the cause of congenital aniridia in a great majority of the cases, but variants in other genes also may lead to aniridia, the FOXC1 and PITX2 genes are among them.
- Epidemiological Data: Providing information about the prevalence of these conditions in Russia and from Orphanet is valuable, but it would be even more informative if the authors could provide epidemiological data on a global scale to better understand the prevalence of these disorders worldwide.
A5: In world population DS prevalence is assessed as 1 per 700 people. (PMID: 35795721) AN prevalence is 1:40,000 to 1:100,000 PMID: 22692063 (Lines 33-34, 47).
- Clinical Characteristics: The description of the characteristic features of both disorders is concise but adequate. However, it may be worth considering adding a more detailed description of the clinical symptoms in congenital aniridia and Down syndrome to help the reader better understand these conditions.
A6: Thank you, added descriptions of DS. Lines 37-42.
- Ocular Anomalies: The description of ocular anomalies associated with Down syndrome is detailed and well-done, but it might be valuable to provide more information about the typical clinical features in congenital aniridia to allow readers to compare both conditions
A7: Ocular anomalies in DS and AN could overlap, strabismus, refractive errors, cataract, and nystagmus occur very often with both syndromes though the frequencies vary. Other affections of the eye could differ, special cornea, iris, optic nerve and retina clinical features characterize each of the two disorders. Thinning and conical shape of the cornea, brushfield spots composed by hyperplasic iris stromal connective tissue, optic nerve head elevation and its vascularization, central macula thickness could characterize DS. While complete or partial iris absence or its severe hypoplasia, cornea thickening, limbal deficiency, keratopathy, optic nerve and foveal hypoplasia are the features of aniridic eye. PMID: 35795721 PMID: 37647922 We added the information in the introduction section, thank you. Lines 48-56
- Study Objective: The study objective is clear - to present a genetically confirmed case of PAX6-associated congenital aniridia co-occurring with Down syndrome in a single patient and to explore potential interactions between these two distinct disorders. This is well-defined and serves as motivation for further reading.
- Overall, the text is well-written and conveys important information. Consideration could be given to providing more detailed clinical symptom data for both conditions and expanding on the epidemiological information.
A9: Thank you for your comment. We have added detailed clinical portrait of both conditions into the Introduction section.
- 2. Case presentation:
- Comments on the Case Presentation:
- Patient Presentation: The description of the patient is detailed and includes important information such as her age, birth conditions, Apgar scores, and the confirmation of trisomy 21 through karyotyping. This is important for providing a comprehensive overview of the patient's situation from birth.
A12: Supplementary Table 1 contains detailed clinical description of the patient in different ages and aspects.
- Child's Development: The description of the child's development focuses on several key developmental milestones, providing insights into her progress and achievements. It's worth noting that these details are useful for assessing her condition and treatment.
- Medical Tests: The authors provide results from various tests, such as neurosonography, eye ultrasound, hearing tests, cardiac ultrasound, and others. This is important for gaining a thorough understanding of the extent of the patient's conditions.
- Eye Findings: The description of eye findings is accurate and includes information about both eyes, which is crucial for evaluating her eye health and vision.
- Clinical Aspects: The authors provide an extensive description of various clinical aspects of the patient, including neurological, ophthalmological, urological, endocrinological, otolaryngological, and orthopedic aspects. This is important to consider the complexity of her health condition.
A16: Thank you. Our aim was to focus on the complex phenotype observing in our patient.
- Medical History: The description of the medical history is comprehensive and includes information on various aspects of the patient's health. This is valuable information for doctors conducting further research and treatment.
- Genetic Test Results: The authors conducted genetic tests and identified a heterozygous variant NM_000280.4:c.282C>A, p.(Cys94*) in the PAX6 gene. They also describe that this variant is not present in the patient's parents and is classified as pathogenic. This is important information that can help understand the cause of her condition.
- Disease Progression: The description of the disease progression of the patient is comprehensive and provides insights into various aspects of her health from birth to the present.
- Discussion:
- Interactions Between DS and AN: In the discussion, the authors address the combination of features associated with Down syndrome (DS) and congenital aniridia (AN). They explain that both conditions may influence each other, but they do not provide a deeper analysis of the molecular or biological mechanisms that might underlie these interactions. It would be valuable to expand this section to understand why these two conditions might co-occur and how they may mutually affect each other.
A21: Thank you, we added a paragraph into the Discussion section. Lines 233--259 contains update of combination of clinical signs of both conditions. Further we added some hypothetical discussions about putative molecular and cellular causes of development and ways to treatment of both conditions.
- Case Analysis: The authors present a comparison of three cases of patients with the combination of DS and AN. However, a more in-depth clinical analysis of each case and their differences and similarities is lacking. Adding a more detailed assessment of the symptoms and test results for each patient would help better understand the distinctions between them.
A22: Table 1 contains the comparison of clinical signs of the three known patients with complex DS and AN. Two previously published have only limited information about clinical picture of the patients. We gathered all available data in the Table 1.
- Potential Pathogenesis: The discussion does not provide an explanation for the pathogenesis of this unique combination of DS and AN. While the authors mention the possibility of genes related to these conditions influencing each other, they do not offer specific information on this topic. This is an important aspect that could be further developed to better understand the reasons for the occurrence of these two conditions in one patient.
A23: We could only suggest the mechanisms, we mention above that there is still a lack of such knowledge. Our very putative suggestion concerns general organism adaptation properties involvement into the survival of the patients with combination of the two severe congenital, chromosome and monogenic, conditions each of which affects embryonal development. Lines 187-195 and 278-282
- Therapeutic Perspectives: Consider adding information about therapeutic perspectives for the patient. Are there any innovative therapies or approaches that could help manage her health condition? This could be valuable for other doctors and researchers dealing with similar cases.
A24: Lines 260-273 are devoted to the discussion of the issue. Review of all possible treatment options for both conditions are out of the scope of the manuscript.
- Summary: The discussion lacks a summary of the main conclusions and messages. Consider adding a brief summary that highlights the key aspects of the discussed case and the insights gained from this analysis.
A25: Thank you. We have significantly updated the discussion section based on the comments. We believe these changes improved the manuscript.
Overall, the discussion is interesting and contains important information about the patient's case, but it could be further expanded and deepened. Particularly, understanding the pathogenesis of the co-occurrence of these two conditions in one individual is crucial.
A26: Thank you very much! We tried to have followed up all the suggestions in comments.
- Conclusions:
- Clinical Assessment: In the conclusions, the authors summarize that the patient has a phenotype that encompasses characteristics of both Down syndrome (DS) and congenital aniridia (AN), with some shared features that do not seem to exacerbate each other. This is a good overall assessment, but it would be beneficial to provide a more detailed description of which features are common and which are different between these two conditions.
A27: Thank you. We have heeded the suggestions and significantly expand Introduction and Discussion sections.
- De Novo Emergence: The authors note that these two conditions appeared de novo in the patient, resulting in a relatively mild phenotype. However, they do not provide a deeper analysis of why and how these two conditions emerged de novo in this specific patient. Exploring the reasons behind this occurrence would be valuable.
A28: In our opinion, the combination of AN and DS does not contradict the possibility of co-occurrence by chance. We have focused on it in Lines 278-282
- Guidance for Parents: The authors mention that parents should remain vigilant regarding potential complications related to the eyes and the immune system in the patient. It would be helpful to add specific recommendations or precautions that parents should take in terms of monitoring and managing these potential complications.
A29: Early stimulation therapy such as therapeutic exercises and behavioral intervention are the most important management. Specific recommendations for the parents we gave in the paragraph Practical recommendations, they concern eye treatment peculiarities. (Lines 260-273)
- Consideration for Future Research: In the conclusions, it might also be worth considering what future research directions could be useful in better understanding this unique case and its clinical implications. Are there genetic or molecular aspects worth delving into in the context of this case?
A30: Co-occurrence of AN and DS could make a subject for investigation of complex genetic interactions between different loci. We could suggest such co-appearance cases might be excellent examples of phenotypic presentations of genetic interaction on genic and chromosomal levels. (Lines 291-294)
- Summary: Consider adding a brief summary that emphasizes the key findings from the analysis of this case and the potential implications for future patient care.
A31: Thank you. We have added a corresponding sentence (Lines 279-283).
Overall, the "Conclusions" section is good but could be enhanced with more detailed information about common and differing features and recommendations for parents regarding monitoring the patient.
A32: Thank you. We have updated the text accordingly.
The overall quality of English language usage in the manuscript is quite good, with clear and understandable writing. However, there are some areas where improvement is needed:
- Grammar and Sentence Structure: There are instances where sentence structure could be more concise and grammatically correct. For example, in some sentences, there is an unnecessary use of passive voice or complex phrasing that can be simplified for clarity.
- Clarity and Precision: While the text is generally clear, there are sections where the meaning of certain sentences or phrases could be clarified for better comprehension. Some sentences appear slightly ambiguous or wordy.
- Terminology and Abbreviations: Ensure that medical and scientific terminology is consistently used and properly defined, especially when introducing abbreviations. Readers should be able to easily understand all specialized terms.
- Consistency: Maintain consistency in formatting, punctuation, and style throughout the manuscript. This includes consistent use of tense (past or present), punctuation (e.g., the use of commas), and formatting (e.g., italics for scientific names).
- Citation and Reference Style: Verify that the citation and reference style adheres to the specific guidelines or standards required by the journal or publication.
- Proofreading: Conduct a thorough proofreading to eliminate typographical errors, such as missing or duplicated words, punctuation mistakes, and formatting issues.
Overall, the manuscript is well-written, but careful editing for grammar, clarity, and consistency will further enhance the quality of the English language usage.
A33: Thank you. We have revised English language throughout the text.
Reviewer 2 Report
In this manuscript, the authors describe the case of a child with both congenital aniridia (AN) as well as Down’s syndrome (DS). The authors share various clinical observations, including visual and other neurological observations, during the first 3 years of the child’s life. Due to such cases being rare, it is useful for the field to have access to such case reports. Comments on the manuscript are:
1. Lines 26-27: The authors mention that Congenital aniridia (AN) results from heterozygous PAX6 variants. This is somewhat misleading because, although Pax6 mutations are the most common cause, there are a few other genes also known to cause congenital aniridia. Additionally, in the Discussion section, the authors describe a patient with two Pax6 mutations, and his father with heterozygous mutation had subtle iris hypoplasia but no congenital anirdia.
2. In the Introduction section or in the Case presentation section, please add a sentence to describe up to what age the patient’s clinical observations were made with respect to the study presented in this manuscript.
3. The authors have described the observation well in the Supplemental Table. In the case presentation section of the main text, it would be better to describe the different clinical studies performed in a chronological order to make it easier to follow.
4. In the Discussion section, please include a brief discussion on what functions of PAX6 are and how the mutation could be causing the phenotype. It does have to be detailed but adding 3-5 sentences on this aspect will improve the Discussion section.
5. Similarly, it would be helpful to include a brief (3-5 sentence) addition to the Discussion section on the range of severity of Down’s syndrome observed in different patients and how it might correlate to this patient’s aniridia and Down’s syndrome diagnosis.
6. Some tracked changes are present in the Supplementary file, please update.
Well-written manuscript but editing is needed to clarify text.
Author Response
In this manuscript, the authors describe the case of a child with both congenital aniridia (AN) as well as Down’s syndrome (DS). The authors share various clinical observations, including visual and other neurological observations, during the first 3 years of the child’s life. Due to such cases being rare, it is useful for the field to have access to such case reports. Comments on the manuscript are:
- Lines 26-27: The authors mention that Congenital aniridia (AN) results from heterozygous PAX6 variants. This is somewhat misleading because, although Pax6 mutations are the most common cause, there are a few other genes also known to cause congenital aniridia. Additionally, in the Discussion section, the authors describe a patient with two Pax6 mutations, and his father with heterozygous mutation had subtle iris hypoplasia but no congenital anirdia.
A1: Thank you, corrected. (Lines 27-28). And, agreed, the father had other than aniridia phenotype.
- In the Introduction section or in the Case presentation section, please add a sentence to describe up to what age the patient’s clinical observations were made with respect to the study presented in this manuscript.
A2: Thank you, we have updated the text accordingly. (Line 57)
- The authors have described the observation well in the Supplemental Table. In the case presentation section of the main text, it would be better to describe the different clinical studies performed in a chronological order to make it easier to follow.
A3: Thank you, we have changed the text based on the ages of the patient.
- In the Discussion section, please include a brief discussion on what functions of PAX6 are and how the mutation could be causing the phenotype. It does have to be detailed but adding 3-5 sentences on this aspect will improve the Discussion section.
A4: Thank you, we have added the description of the PAX6 function in the Discussion section. (Lines 148-154).
- Similarly, it would be helpful to include a brief (3-5 sentence) addition to the Discussion section on the range of severity of Down’s syndrome observed in different patients and how it might correlate to this patient’s aniridia and Down’s syndrome diagnosis.
A5: Thank you, we have added this information to the Discussion section. (Lines 203-207).
- Some tracked changes are present in the Supplementary file, please update.
A6: Thank you, we have corrected Suppl. Table.
Well-written manuscript but editing is needed to clarify text.
Reviewer 3 Report
The authors well presented a unique case of congenital aniridia in a 4-year-old girl diagnosed as Down syndrome at birth. They further analyzed and discussed the combined impact of these conditions on the patient's phenotype. Detailed comments are listed below:
1. In the Case presentation section, please add if there is any hazard exposure to the mother of the patient during the pregnancy, and if the mother went through the regular prenatal screening.
2. In the Case presentation section, please clarify if the first child from the same family presented any abnormality.
3. The title of the Supplementary Table 1., ‘Cown’ should be ‘Down’.
Author Response
The authors well presented a unique case of congenital aniridia in a 4-year-old girl diagnosed as Down syndrome at birth. They further analyzed and discussed the combined impact of these conditions on the patient's phenotype. Detailed comments are listed below:
- In the Case presentation section, please add if there is any hazard exposure to the mother of the patient during the pregnancy, and if the mother went through the regular prenatal screening.
A1: Thank you, corrected (lines 63-67).
- In the Case presentation section, please clarify if the first child from the same family presented any abnormality.
A2: Thank you, clarified (lines 63-67).
- The title of the Supplementary Table 1., ‘Cown’ should be ‘Down’.
A3: Thank you, corrected